# Cyclodextrin-Oligocaprolactone Derivatives—Synthesis and Advanced Structural Characterization by MALDI Mass Spectrometry

**DOI:** 10.3390/polym14071436

**Published:** 2022-03-31

**Authors:** Cristian Peptu, Diana-Andreea Blaj, Mihaela Balan-Porcarasu, Joanna Rydz

**Affiliations:** 1Petru Poni Institute of Macromolecular Chemistry, Grigore Ghica Voda Alley, 41A, 700487 Iasi, Romania; blaj.diana@icmpp.ro (D.-A.B.); mihaela.balan@icmpp.ro (M.B.-P.); 2Polish-Romanian Laboratory ADVAPOL, M. Curie-Skłodowska 34, 41-819 Zabrze, Poland and Aleea Grigore Ghica Voda, 41A, 700487 Iasi, Romania; 3Centre of Polymer and Carbon Materials, Polish Academy of Sciences, M. Curie-Skłodowska 34, 41-819 Zabrze, Poland

**Keywords:** cyclodextrin, ε-caprolactone, cyclodextrin-oligoester, MALDI mass spectrometry, tandem MS, NMR spectroscopy, ring-opening oligomerization, kinetics, biodegradable polyesters

## Abstract

Cyclodextrins have previously been proven to be active in the catalysis of cyclic ester ring-opening reactions, hypothetically in a similar way to lipase-catalyzed reactions. However, the way they act remains unclear. Here, we focus on β-cyclodextrin’s involvement in the synthesis and characterization of β-cyclodextrin-oligocaprolactone (CDCL) products obtained via the organo-catalyzed ring-opening of ε-caprolactone. Previously, bulk or supercritical carbon dioxide polymerizations has led to inhomogeneous products. Our approach consists of solution polymerization (dimethyl sulfoxide and dimethylformamide) to obtain homogeneous CDCL derivatives with four monomer units on average. Oligomerization kinetics, performed by a matrix-assisted laser desorption ionization mass spectrometry (MALDI MS) optimized method in tandem with ^1^H NMR, revealed that monomer conversion occurs in two stages: first, the monomer is rapidly attached to the secondary OH groups of β-cyclodextrin and, secondly, the monomer conversion is slower with attachment to the primary OH groups. MALDI MS was further employed for the measurement of the ring-opening kinetics to establish the influence of the solvents as well as the effect of organocatalysts (4-dimethylaminopyridine and (–)-sparteine). Additionally, the mass spectrometry structural evaluation was further enhanced by fragmentation studies which confirmed the attachment of oligoesters to the cyclodextrin and the cleavage of dimethylformamide amide bonds during the ring-opening process.

## 1. Introduction

Cyclodextrin-polyester derivatives are highly important in the biomedical field as they combine both the properties of cyclodextrins (CDs) and of polyesters such as drug encapsulation capacity, low toxicity, biocompatibility, and biodegradability [1]. Native CDs are poorly water-soluble and the chemical modification of CDs at hydroxyl groups is often employed to improve their solubility through esterification, etherification, amination, etc. [2,3]. Polyester-modified CDs can be prepared through the core-first procedure, the ring-opening reaction of cyclic esters in the presence of CDs that act as initiators, and through the arm-first procedure, the grafting to or from CDs of previously synthesized polyesters. Moreover, CD molecules can be incorporated into various polyester matrices and possibly other polymers, retaining their capacity to form inclusion complexes with different hydrophobic molecules—especially drugs—to increase their solubility. Thus, various formulations (hydrogels, fibers, particles, micelles, etc.) based on CDs and polyesters (generally, poly(ε-caprolactone)—PCL and polylactide) have been prepared for drug delivery applications [4,5,6,7,8,9,10,11,12,13].

Common ring-opening polymerization (ROP) catalysts have been used to prepare well-defined star-shaped polymers by the core-first procedure—for example, tin octoate [12,13,14,15,16] or various amines such as 4-dimethylaminopyridine (DMAP) for lactide [17,18,19] and (–)-sparteine (SP) for β-butyrolactone [20,21]. Organometallic catalysts have also been successfully used for the CD-initiated ROP of ε-caprolactone (ε-CL) [22,23], *L*-lactide [24], lactones, and carbonates [25]. However, the organocatalysts have the advantage of being easily removed (e.g., by washing or trapping in resin beads [26]), and the lack of metal traces, usually encountered when using organometallic catalysts, makes the products suitable for biomedical applications [27].

PCLs have been prepared previously with very good results using alcohol initiators and various organocatalysts, such as acids [28,29,30,31,32,33,34,35] and binary systems comprised of thioureas with different bases [36,37,38,39] or with trifluoroacetic acid [40], as well as mixtures of DMAP and its protonated form with trifluoromethanesulfonic acid [41]. In general, acid organocatalysts are more active for the ring-opening of ε-CL, but when dealing with CDs as initiators, the organocatalysts have to be carefully chosen, as CDs may undergo hydrolysis in acidic media—which increases with temperature and concentration [2]; however, in basic media, CD molecules are stable.

The synthesis of cyclodextrin-polyester derivatives by the ring-opening of cyclic esters can be performed using CD as an initiator, through its many hydroxyl groups, in the presence of classical catalysts (organometallic or organic nucleophilic activators). However, for the bulk ROP of cyclic esters performed in the absence of a cocatalyst, the CD molecules may act as catalysts and initiators, as has already been proven for β-butyrolactone, δ-valerolactone, *L*-lactide, and ε-CL [42,43,44,45]. Monomer activation takes place via encapsulation in the CD cavity and the CD acts as a catalyst and initiator, leading to the formation of star-shaped derivatives. Additionally, it has been hypothesized that the modification of the CD molecule takes place at the hydroxyl at position 2 of the glucopyranose unit [42,43,44]. An attempt to explain the effect of CD on the ring-opening of cyclic esters took into consideration the physical complexation of lactone cycles inside the CD cavity and activation followed by an attack of the activated lactone performed by a secondary hydroxyl group at position 2 of the CD that led to the cleavage of the ester bond and the CD’s modification at this position. However, lactide bulk polymerization, only in the presence of CD, leads to multiple esterifications, selectively at primary hydroxyl groups of CD [45]. Moreover, the modification of CD also takes place at the primary hydroxyl groups, under the conditions of solution ring-opening oligomerization of lactides [19,46,47].

The ring-opening of ε-CL in the presence of β-CD in solution (pyridine [23], *N*,*N*-dimethylformamide (DMF) or dimethylacetamide (DMAc) [22]) using additional catalysts (NaH or yttrium trisphenolate), leads to CD substitution at the primary hydroxyl groups, as determined by NMR experiments; the use of higher NaH quantity leads, however, to CD modification at all positions of the glucopyranose unit. More recently, the bulk ring-opening polymerization of ε-CL was performed in the presence of β-CD (both wet and dry), at atmospheric pressure, and in high-pressure systems and inert media [48]. Using wet β-CD under pressure for ε-CL, the conversion rate increased significantly. However, only a minor fraction of PCL was initiated by the hydroxyl groups of β-CD, the majority instead being initiated water molecules from the reaction system, having hydroxyl or carboxyl end chain groups, as revealed by matrix-assisted laser desorption/ionization mass spectrometry (MALDI MS) analysis. In these high-pressure systems, the CD molecule acts primarily as a catalyst, similar to lipase enzymes. Nevertheless, water-initiated transesterification reactions are the main reason for the formation of PCL homopolymers.

This work aims to optimize the synthesis of biodegradable cyclodextrin-oligocaprolactone derivatives (CDCL). To achieve this, advanced structural characterization is performed for the resulted complex reaction mixtures in order to differentiate modified cyclodextrins with various substitution degrees and patterns. Thus, the ring-opening reaction kinetics were followed both by MALDI MS and ^1^H NMR to confirm the obtained results, and the final product was structurally characterized by ^1^H NMR, ^13^C NMR, and mass spectrometry fragmentation studies.

MALDI MS represents a powerful technique for the characterization of complex chemical structures, allowing the identification of monomer units and end chain groups, and the precise determination of the average molecular weights of polymer samples with narrow dispersity [49,50]. Moreover, MALDI MS may be employed for witnessing minute changes in molecular weights during polymerization reactions—thus expanding its use, as shown by previous studies concerning the ring-opening polymerization (ROP) of lactide [19,51,52]. MALDI MS has already been employed for the in-depth structural characterization of cyclodextrin-oligoester derivatives [17,19,25,42,46,48]. More specifically, MALDI mass spectrometry has previously proven its utility in studying the reaction kinetics of the ring-opening polymerization of lactide, showing an excellent agreement with SEC and ^1^H NMR [19,51].

Here, the ring-opening oligomerization (ROO) of ε-CL initiated by β-CD was thoroughly investigated. The reaction was carried out in solution, aiming to observe the effect of the synthesis solvents (DMF and dimethyl sulfoxide—DMSO) on the structure of the obtained products. Moreover, two nucleophilic organic activators (DMAP and SP), previously used to obtain other CD-polyester derivatives, were employed for the ring-opening process of ε-CL to establish their influence on our particular reaction conditions.

## 2. Materials and Methods

### 2.1. Materials

ε-Caprolactone (ε-CL—Sigma Aldrich, Saint Louis, MO, USA) was distilled under reduced pressure after being dried overnight with CaH_2_. β-Cyclodextrin (β-CD—Cyclolab, Budapest, Hungary) was dried under vacuum at 100 °C for 72 h and kept in the desiccator over P_2_O_5_ under Ar atmosphere. The solvents, *N*,*N*-dimethylformamide (DMF—Sigma-Aldrich, Saint Louis, MO, USA), and dimethyl sulfoxide (DMSO—Sigma-Aldrich, Saint Louis, MO, USA) were distilled under vacuum before use. The catalysts, 4-dimethylaminopyridine (DMAP) and (–)-sparteine (SP) were purchased from Sigma-Aldrich, Saint Louis, MO, USA and used as received. NMR solvent, DMSO-d6 (99.8% D), was acquired from Eurisotop, Gif sur Yvette, France. For MALDI MS analysis, the matrix (2,5-dihydroxybenzoic acid—DHB or *α*-cyano-4-hydroxycinnamic acid—CHCA), the cationization agent (sodium iodide—NaI), and Amberlyst 15 hydrogen were purchased from Sigma Aldrich, Saint Louis, MO, USA, while methanol, acetonitrile, and diethyl ether were purchased from VWR International (Vienna, Austria).

### 2.2. Synthesis

The cyclodextrin-caprolactone (CDCL) derivatives were prepared in solution according to Table 1. All solution reactions were performed at 120 °C, at a β-CD/ε-CL molar ratio of 1/8, the ε-CL concentration being 1.8 M in the solvent (DMF or DMSO). DMAP or SP organocatalysts were added in a 1/1 β-CD/catalyst molar ratio. The reaction kinetics through MALDI MS and ^1^H NMR were followed by collecting fractions for 96 h. CDCL products were obtained by repeated dissolution in methanol and precipitation in cold diethyl ether after trapping the organocatalyst in Amberlyst 15 resin. Afterwards, the products were dried under vacuum at 60 °C and characterized by NMR and MALDI MS.

CDCL product: ^1^H NMR (400.13 MHz, DMSO-d6, δ, ppm): 5.93–5.47 (m, OH-2, OH-3), 5.15–5.08 (m, H-3′), 5.03–4.72 (m, H-1, H-1′), 4.60–4.38 (m, OH-6, OH-ε, H-2′, H6′), 4.13 (H-6′), 4.00–3.98 (chain H-ε), 3.88–3.86 (H-3′, H-5′), 3.64–3.57 (H-3,5,6), 3.39–3.19 (OH-CH_2_-ε, H-2, H-4, overlapped with solvent residual water), 3.37–2.26 (H-α, chain H-α), 1.59–1.49 (H-β, chain H-β, chain H-δ), 1.44–1.38 (H-δ), 1.34–1.28 (H-γ, chain H-γ). ^13^C NMR (100.6 MHz, DMSO-d6, δ, ppm): 172.9–172.6 (O-C=O), 102.5–98.5 (C-1), 82.3–81.3 (C-4), 73.0–68.9 (C-2,2′, C-3,3′, C-5,5′), 63.5 (chain C-ε), 63.2 (C-6′), 60.7–60.5 (OH-C-ε), 59.6 (C-6), 33.9–33.2 (C-α, chain C-α), 33.2–32.1 (C-δ), 28.0–27.8 (chain C-δ), 25.1–24.8 (C-γ, chain C-γ), 24.4–23.8 (C-β, chain C-β).

### 2.3. Characterization

Mass Spectrometry: MALDI MS analysis was performed using RapifleX MALDI TOF TOF MS (Bruker, Bremen, Germany). The FlexControl 4.0 and FlexAnalysis 4.0 software (Bruker, Bremen, Germany). ) were used to control the instrument and process the MS and MS/MS spectra. The collected samples (20 μL) were dissolved in 1 mL of methanol containing Amberlyst 15 and mixed using a Vortex-Genie 2 device. The DHB matrix and NaI solutions were prepared in methanol at concentrations of 20 mg/mL and 5 mg/mL, respectively, while CHCA was prepared in water/acetonitrile mixture (1/1 *v*/*v*). The samples analyzed using the DHB matrix were applied on the MALDI steel plate using the dried droplet method: 20 μL of DHB was mixed with 2 μL of NaI and 2 μL of sample solution, and 1 μL from this mixture was deposited on the ground steel plate. Samples prepared using the CHCA matrix were applied using the thin-layer method: 1 μL of CHCA solution was applied to the target and left to dry, followed by the deposition of 0.5 μL of sample solution on top of the matrix, which was allowed to dry prior to analysis. The spectra were acquired in the positive reflectron mode and the laser ionization power was adjusted just above the threshold to produce consistent MS signals. The “partial sample” shooting mode, which covers a small area around the initial shooting site, was used to collect 18k spectra from different regions of the spot. The obtained MALDI MS spectra were further used to follow the reaction kinetics and to characterize the final product. The MS calibration was performed using poly(ethylene glycol) standards applied to the MALDI target with DHB or CHCA matrices. The MS/MS fragmentation experiments were performed in LIFT mode using a Bruker standard fragmentation method. The full isotopic profile of the parent ion was isolated.

The average molecular weights and the dispersity index were determined by MALDI MS using the following formulae:(1)$$M_{n}=\frac{\sum_{i}^{n}I_{i}*m_{i}}{\sum_{i}^{n}I_{i}}$$

Equation (1)—numeric average molecular weight
(2)$$M_{w}=\frac{\sum_{i}^{n}I_{i}*m_{i}{}^{2}}{\sum_{i}^{n}I_{i}*m_{i}}$$

Equation (2)—gravimetric average molecular weight
(3)$$Đ=\frac{M_{w}}{M_{n}}$$

Equation (3)—dispersity index

Where *I_i_—*monoisotopic peak intensity corresponds to the *m_i_; m_i_*—*m/z* value of the corresponding *i* peak, with *z* = 1.

Peak integration was performed in the flexAnalysis software (Bruker) using the following parameters: snap peak detection algorithm, signal to noise threshold = 6, maximal number of peaks = 100. These settings automatically provide MS peaks lists containing the monoisotopic peak from the isotopic cluster. In the case of overlapping isotopic clusters, the monoisotopic peaks were manually assigned.

Nuclear magnetic resonance (NMR): The 1D and 2D NMR spectra were recorded on a Bruker Avance NEO 400 MHz Spectrometer (Bruker, Rheinstetten, Germany), equipped with a 5 mm probe for the direct detection of H, C, F, Si. All the spectra were recorded at room temperature using DMSO-d6 as a solvent and standard parameter sets provided by Bruker. The chemical shifts are reported as δ values (ppm) relative to the solvent residual peak (2.51 ppm for ^1^H and 39.5 ppm for ^13^C). The ^13^C NMR spectrum was recorded with 16,384 scans and the ^13^C-DEPT135 spectrum was recorded with 6144 scans. The experimental conditions for the 2D experiments were as follows: ^1^H,^1^H-COSY (correlation spectroscopy) pulse program ‘cosygpppqf’, 2048 × 256 data points, ds = 16, ns = 1, spectral width 16 ppm, d1 = 2 sec.; ^1^H,^13^C-HSQC (heteronuclear single quantum coherence) pulse program ʻhsqcetgpsi2ʼ, 1024 × 256 data points, ds = 16, ns = 6, spectral width F1 28 ppm × F2 220 ppm, d1 = 1.5 s. and ^1^H,^13^C-HMBC (heteronuclear multiple bond correlation), pulse program ʻhmbcgplpndqf ʼ, 2048 × 256 data points, ds = 16, ns = 8, spectral width F1 28 ppm × F2 220 ppm, d1 = 1.5 s.

The substitution degree was determined by NMR using the following formula:$$n=\frac{I_{H-\alpha }}{I_{H-1}}\times \frac{7}{2}$$
where *n*—the number of CL units bound to one β-CD molecule; *I_H-α_*—the integral value of the peaks for chain and end-chain H-α (3.37–2.26 ppm); and *I_H-_*_1_—the integral value of the peaks for the anomeric protons of β-CD (5.03–4.72 ppm).

The average chain length of the oligoester chains was calculated using the formula:$$\frac{I_{H-\alpha }}{I_{H-\alpha }-I_{H-\varepsilon \;chain}}$$
where *I_H-α_*—the integral value of the peaks for chain and end-chain H-α (3.37–2.26 ppm) and *I_H-ε_* chain—the integral value of the peak for chain H-ε (4.00–3.98 ppm).

The average number of oligocaprolactone arms per β-CD can be calculated using the formula:$$\frac{7}{2}\times \frac{I_{H-\alpha }-I_{H-\varepsilon \;chain}}{I_{H-1}}$$

For ^1^H NMR reaction kinetics: for each of the samples collected at different reaction times, 50 µL of the reaction mixture was added to 450 µL of DMSO-d6. For optimal homogenization of the solutions, each sample vial was vortexed for 5 min at 500 rpm. The samples were then transferred into NMR tubes for recording the spectra. The reference integral was set on the peak for two of the aromatic protons of DMAP (6.59 ppm). DMAP (of a known concentration) was used as an internal reference for calculating the ε-CL concentrations using the formula:$$C_{\varepsilon -CL=}\frac{I_{\varepsilon -CL}}{I_{DMAP}}\times \frac{N_{DMAP}}{N_{\varepsilon -CL}}\times C_{DMAP}$$
where *I_ε-CL_* is the value of the integral for the −CH_2_ protons from position ε of ε-CL (4.21 ppm); *I_DMAP_* is the value of the integral for the 2 aromatic protons of DMAP; *N_DMAP_* and *N_ε-CL_* are the number of protons from DMAP and ε-CL, respectively, that give the integrated peaks; and *C_DMAP_* is the molar concentration of DMAP (1.895 × 10^−2^ M) in the analyzed solution.

## 3. Results and Discussion

In this work, CDCL derivatives were obtained by the ring-opening reaction of ε-CL in the presence of β-CD and two nucleophilic activators (DMAP and SP; Figure 1). The reactions were performed in solution (DMSO or DMF), at 120 °C, using a CD/catalyst molar ratio of 1/1 and a CD/CL molar ratio of 1/8, according to Table 1. Preliminary analysis revealed that the obtained CDCL derivatives had increased solubility as compared with native CD, e.g., in water (up to 1.7 g/mL), methanol, DMF, and DMSO.

The data provided in Table 1 reveals that structurally homogeneous CDCL products may be obtained in relatively good yields. Previous attempts [42,43] performed in bulk conditions (100 °C), at 1/5 CD/CL molar ratio resulted in poor monomer conversion (15% yield) and, consequently, low CDCL yields—possibly because native cyclodextrins are insoluble in the ε-CL monomer. Such conditions lead to the formation of CDCL and unreacted CD/ε-CL mixtures, which require further purification steps. In a more recent study [48], employing increased pressure conditions, at 120 °C temperature, monomer conversion was also low—around 5%. According to the presented data, employing solution polymerization with additional organocatalysts leads to improved ε-CL conversion.

Here, MALDI mass spectrometry was employed for the evaluation of the ring-opening reaction kinetics of ε-CL in the presence of β-CD to find the optimum conditions for CDCL derivative synthesis, by following the average molecular weight (*M_n_*) evolution. Previously, the influence of different reaction conditions on the ring-opening oligomerization of *D*,*L*-lactide in the presence of β-CD has been studied by MALDI mass spectrometry [19]. Besides the changes in molecular weight, the MALDI mass spectra revealed the presence of secondary products when different solvents and temperatures were employed for the synthesis of cyclodextrin-oligolactide derivatives, whose structure was confirmed by fragmentation studies.

### 3.1. MALDI MS Characterization of CDCL Derivatives

The mass spectrum of a typical CDCL product, obtained through the ring-opening of ε-CL in the presence of *β-CD* and DMAP (#2—Table 1) and using DMSO as the solvent, is presented in Figure 1. The mass spectrum consists of peak series with a 114 Da peak-to-peak difference (corresponding to the 6-oxy-hexanoate constitutional unit noted as CL) starting from the peak with *m/z* = 1157, corresponding to the sodium adduct of *β-CD* (*m/z =* 1134 (*β-CD*) + 23 (*Na^+^*)). Therefore, the main series can be described by *m/z* = 1134 (*β-CD*) + *n**114 (*CL*) + 23 (*Na^+^*). The sample *M_n_* was quantified using the *m/z* ratio and the intensity of the peaks. Thus, the *M_n_* value determined for the typical CDCL product was 1567 g/mol, corresponding to about 3.6 CL constitutional units per β-CD molecule.

MS/MS fragmentation studies were further employed to confirm the structures of CDCL main products. The fragmentation of CDCL derivatives proceeds similarly to other CD-oligoester derivatives [19,20,45,46], having two main pathways: the cleavage of the 1,4-glycosidic bond of β-CD [53,54,55] and the cleavage of the ester bonds from the CL attached to the β-CD molecule [56,57,58]. Oligocaprolactone fragmentation can occur either on the acyl side through 1-3 hydrogen rearrangements (Figure 2A—*a series*) or on the alkyl side through 1-4 hydrogen rearrangements of the ester bond (Figure 2A—*b series*). If the CDCL fragmentation takes place on the acyl side of the ester bonds, the daughter ions are hydroxyl-terminated and the neutral losses correspond to multiples of 114 Da (CL constitutional unit). If the fragmentation occurs on the alkyl side, the neutral losses correspond to carboxyl-terminated CL units (*n**114 + 18) Da, while the daughter ions present a terminal double bond. The fragmentation of neat CDs takes place with the neutral loss of structures with a mass equal to *x**162 Da (*x* takes values from 1 to 6), corresponding to the molecular weight of one or more glycoside units from the CD molecule. For modified CDs, the fragmentation occurs via neutral losses of (*x**162 + *y*) Da—*y* representing the molecular weight of the attached structures; in our particular case, *y* = *n**114, which corresponded to the CL units (Figure 2B). Deeper studies describing the mechanistic processes occurring during this type of cleavage may be found in recent reports [54,55].

The fragmentation mass spectrum of the [CDCL_5_ + Na]^+^ parent ion, having five CL constitutional units, is presented in Figure 2 (highlights) and Appendix A (full spectrum). The MS/MS spectrum revealed the corresponding fragment ions and neutral losses resulting from three distinct fragmentation pathways, the cleavage of the ester bonds (*a*, and *b series*) and the glycoside units (the *c series*). The cleavage on the acyl side of the ester bond (*a series*) led to consecutive neutral losses of 114 Da (the fragment ions are indicated in the MS/MS spectrum from Figure 2), whose structures are presented in Figure 2A—*a series*. The cleavage of the ester bond on the alkyl side led to higher intensity fragment ions (*b series* members indicated in Figure 2), with structural assignments presented in Figure 2A—*b series*.

The *c series* onset with a neutral loss corresponding to the glucose unit, identified in the mass spectrum at *m/z* = 1564.6 (justified by a 162 + *n**114 neutral loss where *n* = 0). The cleavage of unsubstituted glycoside units (neutral losses of *x**162 Da—for fragment c_0_, *x* = 1) was prevalent due to the low substitution degree of the CDCL parent ion. This fragment was accompanied by another series of members resulting from the cleavage of glycoside units substituted with one CL unit at *m/z* = 1450.6 (neutral loss of 162 + *n**114 with *n* = 1—fragment c_1_) or two CL units at 1336.6 (neutral loss of 162 + *n**114 with *n* = 2– fragment c_2_), respectively. Thus, the fragmentation experiment confirms the proposed structure of the CDCL product. The presence of the fragments belonging to the *c series*, having 0, 1, or more CL units demonstrated the random character of the CL attachment to different glycoside units, and that the attachment of a single oligocaprolactone chain should be excluded. However, in order to further investigate the attachment of CL units to various positions on the glycoside ring, e.g., esterification at the OH groups at the 2, 3, or 6 positions, advanced NMR structural characterization was employed (Section 2.2.).

### 3.2. NMR Characterization

Generally, structural characterization of modified CDs aims to reveal the substitution degree, the oligoesters’ arm-length, and their substitution site. In particular, the NMR characterization of partially esterified cyclodextrins represents a difficult task because of the high number of resulting signals, peaks overlapping, and lack of comparable standard substances. Therefore, 1D and 2D NMR techniques (Figure 3, Figure 4 and Figure 5 and Appendix A) were employed to complete the structural elucidation of the CDCL derivatives. Loss of the symmetry of the β-CD molecule and changes in the chemical environment due to substitution lead to crowded spectra with broad and overlapping peaks, making interpretation difficult. The ^1^H NMR spectrum of a CDCL product (#2) shows the peaks for the substituted and unsubstituted glucopyranose units of β-CD, as well as the peaks for the CL units attached to the β-CD (structural assignments in Figure 3, spectrum with peak integrations in Appendix A). On average, the number of CL units bound to one β-CD molecule was 3.5, a close value to that obtained from MALDI MS. Moreover, by comparing the integral value of the signal corresponding to the protons from the ε position of the chain (H-ε) with the one corresponding to the protons from the α position (H-α), it was found that 78% of the CL constitutional units had OH end groups, while 22% were bonded through ester bonds to another CL constitutional unit. In other words, the average chain length of the oligoester chains was found to be 1.28 constitutional units per chain. This small value was expected considering the low number of CL units per β-CD molecule (3.5 CL/β-CD). Taking into account the average number of CL units per chain and the average number of CL units per β-CD, the average number of oligocaprolactone arms per β-CD was found to be 2.8 oligocaprolactone arms per β-CD.

The ^13^C NMR spectrum interpretation confirmed the structural assignment of the CDCL #2 product, as it could be observed from the assignments of the characteristic resonance peaks of the substituted and unsubstituted β-CD carbons and CL units (Figure 4). COSY, HMBC, HSQC and ^13^C-DEPT135, experiments (spectra in Appendix A) were used to establish the chemical structures corresponding to the observed peaks, but unfortunately, some could not be clearly assigned because of peak overlapping in the ^1^H NMR spectrum. The HMBC experiment was also useful in assigning the peaks for the chain or hydroxyl-bound CH_2_-ε groups by showing only one long-range correlation peak between the H-ε protons (from 3.99 ppm) and the carbon atom from the carbonyl group (Appendix A). The peak at 3.3 ppm did not give a correlation with the carbonyl corresponding peaks, meaning that it was bound to hydroxyl groups. Although the peak at 3.3 ppm was overlapping with the water signal, it could be easily assigned based on its correlation peak from the HSQC spectrum with the carbon atom at 60.7–60.5 ppm for C-ε, and also based on its long-range correlation peaks with C-γ and C-δ (Appendix A).

The ^1^H NMR spectra recorded to follow the reaction kinetics (Appendix A) offered some information about the course of the ring-opening process and, more specifically, about the substitution site on β-CD. The collected NMR spectra showed peaks belonging to the starting compounds and the reaction products that could not be fully assigned because of the complex mixture of different structures; nevertheless, some data could be extracted using additional COSY and HSQC experiments. The recorded ^1^H NMR spectrum for the reaction sample collected after 30 min of reaction time (Appendix A) showed that the ring-opening reaction of ε-CL in the presence of β-CD and DMAP, using DMSO as a solvent, had already begun. The COSY experiment evidenced resonance peaks for two differently substituted β-CD glucopyranose units at C-3 and C-2. The C-2 substitution, the structure depicted in Figure 5, had H-2 at 4.42 ppm and the corresponding neighboring H-3 at 3.88 ppm, while the C-3 substitution had the H-3 at 5.13 ppm and the corresponding H-2 at about 3.4 ppm. Moreover, as the reaction time increased, the peaks in the ^1^H NMR spectra became broader.

In the ^1^H NMR spectra for the reaction sample collected after 6 h, the peak corresponding to the methylene from the ε position of the CL residues began to appear at 3.99 ppm. Even if this peak partially overlapped with one of the satellites of the peak for CH_2_-ε of the unreacted ε-CL, it was confirmed by the COSY experiment due to its coupling with chain CH_2_-δ (Appendix A). Moreover, for the sample collected after 11 h of reaction time, the NMR spectrum revealed resonance peaks corresponding to the two H-6′ protons of β-CD glucopyranose units substituted at the hydroxyl groups from position 6. These broad peaks partially overlapped with other peaks from the NMR spectrum, but their presence could be confirmed through their correlation peaks in the HSQC spectrum with C-6′ from 63.2 ppm (Appendix A). As the reaction continued, the peaks for H-6′ increased in intensity and could be clearly observed in the spectrum for the final product (Figure 3) and also in the ^13^C NMR (Figure 4), HSQC, and DEPT135 experiments (Appendix A).

Thus, it may be observed that although the final product was randomly substituted, the ROO process had a different course throughout the total reaction time as the substitution began at the larger rim, both at C-2 or C-3, and that after a certain period, the substitution also occurred at C-6. Previously, our studies revealed in the case of *D*,*L*-lactide that the substitution occurs predominantly at C-6; however, the reaction conditions were different: DMF solvent and no DMAP catalyst. Nevertheless, the different substitution patterns between these two systems are rather difficult to explain.

### 3.3. MS and NMR Kinetics

MALDI MS characterization allowed for the precise evaluation of various reaction systems. The ring-opening kinetics of ε-CL using β-CD as an initiator and DMAP as an organocatalyst (#2) were followed by both MALDI mass spectrometry using 2,5-dihydroxybenzoic acid or α-cyano-4-hydroxycinnamic acid as a matrix for sample preparation, and ^1^H NMR spectroscopy to confirm the obtained results. Thus, MS results were employed for the quantification of sample *M_n_* and consequently the monomer conversion evolution, assuming that all the reacted ε-CL was transformed into a CDCL product, without any secondary reactions such as H_2_O initiation (as, e.g., in the work of Galia et al. [48]). On the other hand, the concentration of unreacted ε-CL was calculated from ^1^H NMR spectra (as described in Section 2.3) in order to directly obtain the ε-CL conversion. Sample fractions were collected from the reaction mixture at specific times, inactivated (by cooling and basic activators removal), and analyzed by NMR and MS to determine the monomer conversion.

The comparison between the two methods was performed because MALDI MS determination of CD-oligoesters *M_n_* may be biased if the method is not optimized for a particular polymer analysis; thus, it required confirmation of the results by other techniques such as NMR spectroscopy [59]. The various causes for such bias are related to the sample preparation (choice of matrix, sample solubility and concentration in different solvents, cationization agents, and deposition technique), instrument settings (especially the laser power), and molecular weight distribution. The matrix is one of the most important parameters for MALDI MS analyzes and is usually chosen based on trial and error processes [50].

The MALDI mass spectrometry characterization of PCLs was mostly performed using DHB as a matrix [32,35,40,60,61,62,63], but other matrices were also employed: dithranol [33,50,64], 2-(4-hydroxyphenylazo)-benzoic acid [57], and CHCA [39]. For CD derivatives, CHCA [25,45,46,65,66,67,68,69] and DHB [70,71,72] were the most employed matrices; therefore, we compared the *M_n_* evolution using these two matrices for MALDI MS analysis (Figure 6). The samples analyzed with CHCA were applied to the MALDI target using the thin-layer technique, as the common dried-droplet method can lead to overestimated values [69]. However, in the case of CDCL, we observed that CHCA still led to slightly overestimated *M_n_* values as compared with DHB, while for cyclodextrin-oligolactide type oligoesters, significant differences between these two matrices were not observed [19]. The differences in the case of CDCL derivatives could also be a consequence of the different solubilities of the samples in methanol and the water/acetonitrile mixture used for matrix preparation.

The evolution of *M_n_*_NMR_ and *M_n_*_MALDI_ obtained using both DHB and CHCA matrices is presented in Figure 6. An excellent MS-NMR agreement was observed using DHB as the MALDI MS matrix, with a similar *M_n_* evolution—reaching 1600 g/mol at the end of the reaction time, which corresponds to 3.85 CL constitutional units per β-CD molecule. Moreover, the plot of the *M_n_* values determined through both MALDI MS using DHB and ^1^H NMR, was characterized by a linear evolution with y = x and R^2^ = 0.997 (Appendix A). When CHCA was employed for the MS analysis, the obtained *M_n_* values were consistently higher than those obtained by NMR, as previously noticed in the case of hydroxypropyl-β-CD derivatives [66]. At the end of the reaction, *M_n_* reached 1665 g/mol with CHCA, corresponding to about 4.5 CL units per β-CD. However, when plotting the *M_n_* values obtained with CHCA and NMR (Appendix A), the correlation degree was relatively smaller (y = 1.033×, R^2^ = 0.992) as compared with the one obtained for DHB. Therefore, further MALDI MS analyses were performed using DHB as a matrix.

An important element to be considered was the profile of the observed monomer consumption speed. There were two obvious regions (Figure 6): the first, up to 12 h, was characterized by a fast conversion, and 2.3 CL constitutional units were attached to the β-CD. For the second region, from 12 to 96 h, the conversion was much slower, with an increase in *M_n_* of only 1.55 CL units. Additionally, the NMR analysis revealed that the substitution site on β-CD changed throughout the reaction; in the first phase—which coincided with the fast monomer conversion—the substitution occurred at C-2 and C-3, while in a second phase, there could be observed the esterification, also occurring at C-6.

A possible explanation for the observed *M_n_* evolution relates to a decrease in the overall initiator reactivity towards the ε-CL monomer. The previously observed kinetics in the case of the *D*,*L*-lactide [19] displayed a similar profile, and the decrease in reactivity was related to the formation of secondary OH groups. However, this is no longer true in the case of ε-CL oligomerization. Most probably, in both cases, such a decrease in the reactivity may be associated with β-CD modification that leads to a decrease of cyclodextrin’s capacity for ring-opening the cyclic esters. Thus, once the active sites of native β-CD have been sterically hindered through esterification, the ring-opening reaction no longer benefits from the possible monomer activation via complexation inside the β-CD cavity. This assumption is also based on previous observations of ε-CL conversion limitation in the presence of dry β-CD in bulk [44] or supercritical CO_2_ solvent [48]. However, when water was introduced into the system [48], transesterification reactions occurred with functionalized β-CD, and the oligoesters were transferred from CDCL to linear PCL homopolymers, which finally led to high monomer conversion.

Recent studies [73] have revealed that for lactide monomers, the ROP process (using DMAP catalyst) may proceed without the inclusion of monomer in the CD cavity. The ROP process is accompanied by the formation of non-specific complexes with the monomer physically associated with the outer OH groups of the CD—thus explaining the substitution at primary OH groups from the smaller rim of CD. On the other hand, the ring-opening of CL depends more on the complexation processes in the CD cavity. NMR and MS kinetics revealed that substitution at OH2 and 3 is favored, while substitution at OH6 is slower—probably occurring through transesterification reactions. Therefore, we may infer that specific behaviors may be justified by the particular complexation conditions of CL in β-CD.

### 3.4. Solvent Influence—MALDI MS

In principle, good solvents for both CD and CL may improve the overall system reactivity, allowing for a better diffusion of active species and the formation of homogeneous CDCL species, as compared with bulk polymerization. However, the additional solvent diminishes the concentration of active species and may also favor the occurrence of secondary reactions. In a first approach, we aimed to understand the solvent effect over the ROO process by performing MALDI MS kinetics on two reaction systems in DMSO and DMF, good solvents for both the reagents and products, without additional organocatalysts—#1 and #4 (Table 1), respectively. The monomer conversion evolution presented in Figure 7 reveals that the CDCL product may be obtained in DMSO with a maximal average substitution degree (SD) of 2.13 CL units per β-CD molecule, while in DMF the maximal average SD was significantly higher (4.4 CL/β-CD). The observed ε-CL conversion in DMF occurred in two stages: an initial faster stage up to 12 h followed by a slower increase for the remaining reaction time. On the other hand, in DMSO, there may be observed an initial fast *M_n_* increase, followed by a plateau region. The *M_n_* evolution of the ROO process had a similar profile to the one observed in the case of cyclodextrin-oligolactides [19]: a fast monomer conversion in the initial reaction stage followed by a significantly slower monomer conversion. The higher reactivity of the system using DMF has been previously explained by the occurrence of DMF degradation due to temperature and light. Thus, dimethylamine resulting from DMF degradation acted as an organocatalyst in the ring-opening process.

The products issued from the degradation of DMF were revealed by mass spectrometry analysis. The comparison between MS results obtained for the products synthesized in DMF (#4) and DMSO (#1; Figure 8), showed that besides the main series of peaks presented in both spectra (*m/z* = 1134 (*β-CD*) + *n**114 (*CL*) +23 (*Na*^+^)) belonging to the main CDCL product, a secondary series was presented in the MS spectrum obtained for #4, described by *m/z* = 1134 (*β-CD*) + *n**114 (*CL*) + *m**28 (*formate*) + 23 (*Na*^+^), where *m* represents the number of formate moieties attached to the CDCL derivatives. The MS spectrum of CDCL #4 shows that the *m* values ranged from 0 to 2. The formation of the CDCL secondary products and their structure are described in Figure 3.

DMF degradation interferes with the ROO process and leads to CDCL-formate (CDCL-F) derivatives, which were structurally identified by MS. CDCL-F species result from a complementary process during the nucleophilic attack performed by the hydroxyl groups on ε-CL and leads to the cleavage of the amide bond from DMF, especially at elevated temperatures (Figure 3). The role of DMF as a reagent in the ring-opening of cyclic esters was previously observed only for *D*,*L*-lactide [19], although many other types of reactions may be counted [74]. The traces of dimethylamine resulting from DMF degradation can act as an additional organocatalyst, increasing the ring-opening reaction rate, and thus explaining the higher rate as compared with DMSO.

CDCL-F species were present in low amounts, below the NMR detection capacity in the present product mixture. However, the formate moiety attachment to the CDCL derivatives was further confirmed using MS/MS fragmentation studies. Precursor ions containing five CL constitutional units and one formate moiety [CDCL_5_-F_1_ + Na]^+^ were chosen for fragmentation studies of the CDCL secondary products—Figure 9 and Appendix A. The fragmentation occurred similarly to the CDCL main derivatives, with two main pathways that involved the glycoside and ester bonds. However, the structures of the neutral losses resulting from the cleavage of the ester bonds were specific to the CDCL-F derivatives. When the cleavage of the formate end group took place on the acyl side (Figure 4A—*d series*), the process resulted in a neutral loss of 28 Da, which corresponds to a CO moiety, while for the cleavage on the alkyl side (Figure 4A—*e series*), the observed neutral loss was 46 Da—corresponding to the elimination of formic acid. Moreover, the fragment ions corresponding to the cleavage of the ester bonds, containing CL units with formate end chains, were assigned following the neutral losses of 142 Da (114 (CL) + 28 (CO)) on the acyl side, or 160 Da (114 (CL) + 18 (H_2_O) + 28 (CO)) on the alkyl side; hypothetical structures are presented in Figure 4A (*d* and *e series* of fragments). Additionally, the cleavage of the glycoside bond may be identified in the MS/MS spectra, associated with losses of 162 Da which partially overlap with the 160 Da neutral losses—Figure 9. The neutral loss of glycoside units modified with one CL and one formate unit (304 Da) occurred with the formation of the fragment ion found at *m/z* = 1450.6 (Figure 4B). Thus, it may be observed that both fragmentation processes confirmed the presence of the formate moieties in the structure of the CDCL-F derivatives and supported the hypothesized structural assignment of the parent ions. The fragmentation of the ion species containing two formate moieties, [CDCL_5_-F_2_ + Na]^+^, presented in Appendix A, underwent a similar pattern. The observed main fragment ion, based on its relative intensity, resulted from the cleavage of two formate moieties, respectively, from the acyl side (neutral loss of 56 Da). The cleavage from the alkyl side proceeded via the neutral loss of one or two formic acid moieties (74 and 92 Da, respectively).

### 3.5. Catalyst Influence—MALDI MS

Furthermore, the effect of organocatalysts on the *M_n_* increase of CDCL was investigated by MALDI MS in both the employed solvents: DMSO and DMF. The kinetics plots of the reactions catalyzed by DMAP and SP were compared with the reaction systems without additional catalysts (Figure 10 and Figure 11). DMAP is commonly used with primary or secondary alcohol initiators—β-CD being a suitable compound from this point of view as it has 7 primary and 14 secondary hydroxyl groups, and activates both the initiator and the monomer via a nucleophilic pathway, as previously stated [75,76,77]. On the other hand, SP has been employed previously for the ring-opening polymerization of lactide when it has been used as a cocatalyst with thiourea or fluorinated tertiary alcohols [78,79,80].

From Figure 10, shows that, for DMSO reactions, the measured *M_n_* evolution followed the order SP > DMAP > no catalyst (#3, 2, and 1, respectively). None of the reactions reached the maximal theoretical conversion and the SP catalyzed reaction suffered *M_n_* reduction, possibly because of backbiting reactions. The SP-containing reaction in DMSO also had a clear two-stage evolution, attaining a *M_n_* maximal value of 1620 g/mol at 12 h. After this period, the depolymerization reaction started, decreasing the *M_n_* values and reaching a plateau by the end of the reaction time.

The ring-opening of ε-CL in DMSO without additional organocatalysts is mostly driven by temperature, and β-CD may act as both an initiator, through its hydroxyl groups, and as a catalyst, by forming an inclusion complex with the monomer—similar to the bulk ROP process [42,43,44,48]. From the performed reaction kinetics we may observe that this reaction had the lowest ε-CL conversion, with *M_n_* reaching only 1400 g/mol. However, even in the SP reaction system, which is a strong nucleophilic activator, the monomer conversion was not complete when using DMSO as the solvent. This evolution was a consequence of the high activity of this nucleophilic activator. In the DMAP/DMSO system, after an initial rapid increase, the *M_n_* values were growing slowly and continuously, reaching 1600 g/mol at the end of the reaction time. DMAP addition in DMSO significantly increased the *M_n_* values, as already observed for cyclodextrin-oligolactide derivatives prepared in similar conditions [19].

The kinetics analysis of the DMF reactions revealed a similar behavior (Figure 11), with the same order SP > DMAP > no catalyst (#6, 5, 4). However, it may be remarked that DMF reactions systems are clearly more active than those in DMSO, most probably because of the previously described DMF degradation processes which lead to the formation of the dimethylamine nucleophile activator.

The reactions in DMF without organocatalyst (#4) and with DMAP (#5) had a similar two-stage *M_n_* evolution, with a faster increase for the first 12 h and a slower increase of *M_n_* values for the rest of the reaction time—but, as expected, DMAP led to higher monomer conversion, reaching 1775 g/mol, while DMF alone only led to 1650 g/mol. On the other hand, the addition of SP led to a more distinct two-stage *M_n_* evolution: at the beginning, the reaction took place very fast, with *M_n_* reaching a peak value at 12 h (1845 g/mol); afterward, the depolymerization reaction started, leading to a continuous decrease of *M_n_* for the rest of the reaction time. The system containing SP led in 4 h to the same *M_n_* value as the one with DMAP at the end of the reaction time, but the dispersity obtained using DMAP was significantly lower—Appendix A. The higher dispersity of the average molecular weights may be produced by the increased transesterification reactions in the presence of SP.

## 4. Conclusions

The solution ring-opening oligomerization of ε-caprolactone in the presence of β-CD leads to the formation of highly water-soluble cyclodextrin derivatives. The MALDI mass spectrometry characterization technique, using DHB as a matrix, provided accurate results which were in excellent correlation with the reaction kinetics determined via NMR spectroscopy. NMR characterization of the final CDCL products revealed a random substitution, at the 2, 3, and 6 positions on the glycoside rings. However, the kinetics analysis showed that, initially, a rapid CL attachment occurred at secondary OH groups at the 2 and 3 positions, followed by a slower attachment at primary OH groups at position 6. The decrease in the system reactivity was motivated by the structural modification of the cyclodextrin, which interfered with the role of CD as a catalyst in ROP through the physical inclusion of the monomer. MALDI MS kinetics evidenced that the DMF system was more reactive than DMSO, with and without organocatalysts, because of amide cleavage processes that occurred during oligomerization and the formation of dimethylamine, which in its turn acted as a supplementary organocatalyst. These secondary reactions were revealed by the chemical modification of the CDCL; thus, CDCL-formate derivatives were evidenced using MALDI MS and MS/MS fragmentation experiments. The highest monomer conversion values (approximately 75%) were obtained for the SP organocatalyst, but close values were also reached using DMAP, after a longer reaction time. Besides showing the ROO activity of various reaction systems, MALDI MS kinetics brought indirect proof of backbiting reactions. Overall, mass spectrometry and NMR characterization allowed a good evaluation of the cyclodextrin reaction systems. Further studies will aim to better comprehend cyclodextrin’s involvement in the ring-opening of cyclic esters.

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
