# Peer review of "Cyclodextrin-Oligocaprolactone Derivatives—Synthesis and Advanced Structural Characterization by MALDI Mass Spectrometry"

_polymers, 2022, doi:10.3390/polym14071436_

Round 1

Reviewer 1 Report

The present paper entitled “Cyclodextrin-oligocaprolactone derivatives – synthesis and advanced structural characterization by MALDI mass spectrometry” by the authors Cristian Peptu, Diana-Andreea Blaj, Mihaela Balan-Porcarasu and Joanna Rydz, describes the synthesis of beta-cyclodextrin-oligocaprolactone (CDCL) derivatives via organocatalyzed solution ring opening oligomerization of epsilon-caprolactone in the presence of beta-cyclodextrin. CDCL derivatives were characterized by mass spectrometry fragmentation studies and NMR analysis. MALDI MS was employed for the evaluation of the monomer conversion evolution and the results were confirmed by 1H NMR spectroscopy. MALDI MS was also used for studying the influence of the solvents as well as the effect of the organocatalysts in the ring opening oligomerization process.

After careful evaluation of the manuscript, I consider that it would typically appeal to the readers of Polymers, and therefore I would recommend this paper for publication. The paper has been well-written and organized and it addresses an interesting topic for the scientific community.

My only concern is related with the progress of the substitution reaction in cyclodextrin which is the opposite of what it could be expected. It is well-known that OH-6 groups are the most accessible and most reactive groups in beta-CD; however, NMR seems indicate that this substitution take place after C-2 and C-3 have been substituted even though those groups in position 3 are the least reactive and least accessible in the molecule. I think it would be reasonable to find an explanation for this unusual behavior.

I would also like to suggest that authors should include in the supporting information, an 1H NMR spectrum from CDCL product but including integrals and the calculation they have performed for obtaining the average chain length of the oligoester chains. In this sense, I would like to point out that in the case of H-e chain is difficult to obtain a precise integral value due to its overlapping with other neighboring signals (H-3’, H-5’ and H-6’). Maybe it would be more useful to use a deconvoluted spectrum for a more precise calculation.

Reviewer 2 Report

Dear author,

This paper deals with cyclodextrin-mediated oligocaprolactone synthesis and its structural characterization by the MALDI technique. This is a well written manuscript with interesting information. But there are several matters that must be attended to before being considered for publication.

Abstract section. It must be improved and must contain quantitative data.

Methodology section. Provide more information of NMR heteronuclear experimental conditions used in this work.

Results and discussion section.

Comparison of your results of Molecular weight, yield and other parameters, fractionation patterns, etc.,  with the one previously obtained by other authors to improve the discussion. This comment is valid for the whole manuscript.

Table 1. The Maldi obtained average molecular weight is Mn or Mw? it must be clarified. What is D MALDI and how it was calculated?. Complete the yield column in Table 1.

Lines 285 to 291... must be moved to the Methodology section and placed as a formula. 

Author Response

Answer to reviewers
We would like to express our gratitude to the reviewers and to acknowledge their effort to
improve the paper's quality. Manuscript changes were highlighted in yellow (pdf file).

Reviewer 1

Point 1:

My only concern is related with the progress of the substitution reaction in cyclodextrin
which is the opposite of what it could be expected. It is well-known that OH-6 groups are the
most accessible and most reactive groups in beta-CD; however, NMR seems indicate that
this substitution take place after C-2 and C-3 have been substituted even though those groups
in position 3 are the least reactive and least accessible in the molecule. I think it would be
reasonable to find an explanation for this unusual behavior.

Response 1:

Indeed, the substitution pattern of the CD in the ROO of ε-caprolactone represents an important
issue related to the mechanism of the reaction. Previous papers hypothesized that substitution of
β-CD occurs at C2 position however the arguments supporting such substitution (NMR
characterization) were not accurate. In principle, the formation of ester bonds starting from the
cyclodextrin OH groups may be proved by the disappearance of the OH groups signal and, also,
by the shift of the protons from the C2, C3, and C6, respectively. A selective substitution to one of
these positions would lead to the appearance of novel resonance peaks, as described in the current
paper. However, previous papers failed in bringing such argumentation. Moreover, the reasoning
which was employed to justify a C2 substitution was based on the fact that on β-CD molecule there
are three types of OH groups with their specificities, like OH6 groups are the most accessible for
bulky substituents, the OH2 are the most acidic and OH3 groups are the least reactive

Also, there should be considered that such phenomena occurred in bulk polymerization, without
any interference from the solvent. In our work, the reaction takes place in the presence of solvents
and DMAP. The determined reaction kinetics revealed an overall low reactivity of the OH groups
towards ring-opening of ε-caprolactone and it is possible that CD inclusion complexes are formed
before monomer ring opening. As observed in the early reaction stages, the confined monomer
may favor such a specific substitution (at C2 and C3). We may speculate that the C2 and C3 OHs
should be more sterically available while the C6 OH groups are rather hidden (such hypothesis
may be supported by modeling but at the moment we lack the possibility to test this).

We agree that OH6 are more sterically available. Our work with other cyclic esters, D,L-LA, and
L-LA revealed that substitution takes place at OH 6, rather selectively [45]. On the other hand, a
recent publication [73] revealed that for lactide monomers, the ROP process (using DMAP
catalyst) may proceed without the inclusion of monomers in the CD cavity. The ROP process is
accompanied by the formation of non-specific complexes having the monomer physically
associated with the outer OH groups of the CD.

Thus, we may infer that the observed substitution at OH 2 and 3 is related to the complex formation
and CL ring-opening in the presence of OH groups from the lower rim. In other words, ring-
opening - RO of LA may not necessarily require complexation while RO of CL is depending more
on the complexation. As observed from NMR and MS kinetics, substitution at OH2 and 3 is favored
while substitution at OH6 is slower, occurring probably through transesterification reactions.

The manuscript was modified lines 462-471.

Point 2:

I would also like to suggest that authors should include in the supporting information, an 1H
NMR spectrum from CDCL product but including integrals and the calculation they have
performed for obtaining the average chain length of the oligoester chains. In this sense, I
would like to point out that in the case of H-e chain is difficult to obtain a precise integral
value due to its overlapping with other neighboring signals (H-3’, H-5’ and H-6’). Maybe it
would be more useful to use a deconvoluted spectrum for a more precise calculation.

Response 2:

The figure (S2) was added and the formulae used for calculation were added in the main
manuscript. Indeed, peak overlapping occurs and we performed calculations using deconvolution,
however, without significant differences.
